# Induction of Phlorotannins and Gene Expression in the Brown Macroalga *Fucus vesiculosus* in Response to the Herbivore *Littorina littorea*

**DOI:** 10.3390/md19040185

**Published:** 2021-03-26

**Authors:** Creis Bendelac Emeline, Delage Ludovic, Vallet Laurent, Leblanc Catherine, Inken Kruse, Ar Gall Erwan, Weinberger Florian, Potin Philippe

**Affiliations:** 1Integrative Biology of Marine Models (LBI2M), CNRS, Sorbonne Université, UMR 8227, Station Biologique, Place Georges Teissier, 29680 Roscoff, Brittany, France; ecreis@sb-roscoff.fr (C.B.E.); delage@sb-roscoff.fr (D.L.); laurent.vallet@capgemini.com (V.L.); leblanc@sb-roscoff.fr (L.C.); 2International Research Laboratory IRL 3614, CNRS, Sorbonne Université, PUC, UACH, Evolutionary Biology and Ecology of Algae, EBEA, Station Biologique, 29680 Roscoff, Brittany, France; 3Helmholtz Centre for Ocean Research (GEOMAR), Düsternbrooker Weg 20, 24105 Kiel, Germany; ikruse@kms.uni-kiel.de (I.K.); fweinberger@geomar.de (W.F.); 4Laboratoire des Sciences de l’Environnement Marin, UBO European Institute for Marine Studies IUEM, University of Brest—Western Brittany, UMR 6539 LEMAR, Technopôle Brest Iroise, Rue Dumont d’Urville, 29280 Plouzané, Brittany, France

**Keywords:** phlorotannins, inducible defense, gene expression, grazing, *Fucus vesiculosus*

## Abstract

Mechanisms related to the induction of phlorotannin biosynthesis in marine brown algae remain poorly known. Several studies undertaken on fucoid species have shown that phlorotannins accumulate in the algae for several days or weeks after being exposed to grazing, and this is measured by direct quantification of soluble phenolic compounds. In order to investigate earlier inducible responses involved in phlorotannin metabolism, *Fucus vesiculosus* was studied between 6 and 72 h of grazing by the sea snail *Littorina littorea*. In this study, the quantification of soluble phenolic compounds was complemented by a Quantitative real-time PCR (qRT-PCR) approach applied on genes that are potentially involved in either the phlorotannin metabolism or stress responses. Soluble phlorotannin levels remained stable during the kinetics and increased significantly only after 12 h in the presence of grazers, compared to the control, before decreasing to the initial steady state for the rest of the kinetics. Under grazing conditions, the expression of *vbpo*, *cyp450* and *ast6* genes was upregulated, respectively, at 6 h, 12 h and 24 h, and *cyp450* gene was downregulated after 72 h. Interestingly, the *pksIII* gene involved in the synthesis of phloroglucinol was overexpressed under grazing conditions after 24 h and 72 h. This study supports the hypothesis that phlorotannins are able to provide an inducible chemical defense under grazing activity, which is regulated at different stages of the stress response.

## 1. Introduction

Phlorotannins are polyphenolic compounds that occur exclusively in brown algae. These compounds display a large variety of chemical structures based on the phloroglucinol monomer (1,3,5-trihydroxybenzene), which can be linked to complex polymeric forms described as fucols, phlorethols, fucophlorethols, fuhalols, carmalols and eckols depending on the type of aryl-aryl or diaryl-ether bonds occurring between the monomers [1]. Phloroglucinol is biosynthesized through the acetate-malonate pathway, and produced via the condensation of malonyl-CoA units catalyzed by a type III Polyketide synthase (PKSIII) [2]. Despite the lack of knowledge on the other intermediate steps of their biosynthetic pathway, previous studies underlined the role of phlorotannins in primary and secondary metabolisms [3,4]. Phlorotannins are present in brown seaweed in a soluble form located in cell vesicles named physodes, as described in the zygote of *Hormosira banksii* [5]. These vesicles can be secreted into the extracellular compartment and, through oxidative cross-linking, can form insoluble cell-wall material embedded in the alginate matrix and other cell wall components [6]. These molecules play a key-role in the formation of cell walls at the primary developmental stage of the fucoid zygote [7,8]. Polyphenol molecules carry numerous hydroxyl groups which facilitate the adsorption and adhesion of zygotes to the substrate because they form a rigid adhesive structure by cross-linking cell wall polysaccharides [9].

Phlorotannins produced by brown algae are also involved in multiple stress responses and reveal specific roles in defense mechanisms [1,10]. These compounds present antioxidant and antibacterial activities [11] and take part in the detoxification of Reactive Oxygen Species (ROS) [12]. They also expose a photoprotective role, since they are able to absorb UV radiation when located in the cell wall [13]. Moreover, phlorotannins can deter grazers [14], due to their ability to precipitate proteins [15] which reduces the digestibility of seaweed tissues and interferes with the nutrient acquisition by herbivores [16]. However, mechanisms driving this interaction are still not completely understood. Coleman et al. [17] showed that applying an α-amylase from the saliva of a grazer on *Ascophyllum nodosum* induced the production of phlorotannins after 2 weeks. When studying the effect of grazing on the kelp *Ecklonia radiata*, microscopical observations showed that phlorotannin accumulation started on day 1 and was straightforward from day 3 [18]. Yates and Peckol [19] have also reported that phlorotannin levels in *Fucus vesiculosus* exposed to grazing increased after only 3 days. However, the repelling effect of phlorotannins is still subject to controversy. Indeed, Deal et al. [20] have shown, through a bioassay-guided separation, that galactolipids, rather than phlorotannins, are liable to deter herbivorous sea urchins.

Because constitutive defenses may be very costly for organisms, specific inducible defenses deployed when needed can be more profitable in terms of energy expenses [21]. Such dynamic defenses have also been observed when studying the interaction between the bladder wrack *F. vesiculosus* and the herbivorous periwinkle *L. littorea*. During this interaction, rapid and efficient specific responses described by several short pulses, from a few hours to a whole day, appear as the strategy to optimize defenses [22,23,24,25,26]. In this context, our study attempts to use two complementary methodologies in order to characterize the early inducible responses of *F. vesiculosus* to *L. littorea* grazing. Changes in the soluble phlorotannin content within *F. vesiculosus*, triggered by the grazing activity of *L. littorea*, were investigated over a period of three weeks. In addition, the relative expression of target genes involved in stress and grazing responses as well as phlorotannin metabolism was also monitored during the first 72 h of algae–grazer interactions.

## 2. Results

### 2.1. Phenol Contents

The mean content of soluble phlorotannins obtained in methanolic extracts (*n* = 5) ranged from 18 to 26 mg.g^−1^ DW (Figure 1). A significant effect of grazing was detected over time (Anova *p* value < 0.1) with an increase of 7 mg.g^−1^ DW (ca. 35%) reached within 12 h of exposure to the grazers, compared to the control (estimated marginal means, *p* value < 0.05). However, this content dropped down to the initial steady state at 24 h, and remained constant until the end of the trial.

### 2.2. Gene Expression by qRT-PCR

The study of relative gene expressions selected through previous studies [25,27,28] focused on four main categories of genes: (1) genes known to be related to stress responses, such as heat shock protein 70 (HSP70) and cytochrome p450 (CYP 450), (2) genes encoding enzymes involved in the chemical modification of molecules such as vanadium bromoperoxidase (vBPO) and aryl sulfotransferase (AST6), (3) a gene encoding the polyketide synthase (PKSIII), an enzyme described in the biosynthesis of phlorotannins, and (4) a gene encoding the protein X22181, which is annotated as an outer membrane assembly lipoprotein shown to be significantly upregulated in response to grazing in a recent microarray study (A. Ehlers and I. Kruse, unpublished data), and which could hence have a potential role in maintaining cell wall integrity.

Grazing activity between day 1 and day 3 did not significantly affect the expression of *hsp70* gene (Figure 2a). On the other hand, we observed a significant regulation of the *cyp450* gene expression over time, with a 3.95-fold up-regulation after 12 h grazing (*p* value < 0.01) and a 4.19-fold down-regulation after 72 h grazing (*p* value < 0.01), compared to control condition (Figure 2b).

In the presence of grazers, the *vbpo* gene (Figure 2c) was upregulated by 2.37-folds (*p* value = 0.022) after only 6 h. The relative expression of this gene was nonetheless low compared to the other genes investigated in our study.

After 24 h, a set of differential gene expressions was revealed under grazing conditions. The *ast6* gene (Figure 2d) was overexpressed 2.8 times (*p* value = 0.03), the *pk*sIII gene (Figure 2e) was overexpressed 2.3 times (*p* value = 0.024) and the *x22181* gene (Figure 2f) was also overexpressed 3.8 times (*p* value < 0.01). Later, after 72 h grazing, the *pksIII* gene was overexpressed 1.95 times (*p* value = 0.028) and the *x22181* gene 3.15 times (*p* value = 0.035).

## 3. Discussion

This study aimed to characterize the early effects of grazing on the metabolism of phlorotannin in the brown seaweed *F. vesiculosus*, by using two complementary approaches, direct quantification of soluble phenolic compounds and gene expression.

### 3.1. Quantification of Soluble Phenolic Compounds

Results showed that the concentration of soluble phlorotannin in *F. vesiculosus* increased by 35% within 12 h of exposure to grazing which reveals an obvious induction of phlorotannin production. When periwinkles graze on seaweed surfaces, such as *F. vesiculosus*, only certain zones are damaged. However, in our experiment, the whole thallus was sampled for extracting phlorotannins. Therefore, our results provide an overall concentration level in the tissue without differentiating between systemic and more local physiological responses. The latter could in theory occur with different kinetics in different tissue parts of the alga. Whilst systemic responses toward grazing have been highlighted in vascular green plants [29,30], systemy in brown seaweeds on the other hand has hardly been investigated [31].

The induction of soluble phlorotannin production was quick and transient, which suggests that the algae rapidly sensed the presence of the herbivore or its activity. This first response was then followed by de novo synthesis of phenols which role could either be to reinforce the cell walls, rebuild the exhausted stocks of phenol compounds or to repel the herbivores. Hammerstrom et al. [26] also showed a rapid induction of phlorotannins within between 1 and 3 days after mechanical wounding of four kelp species. In our study, the soluble phlorotannin content did not significantly change after the 12-h peak, thereby suggesting that after 24 h, either the initially synthesized new material reached sufficient amounts for achieving the targeted responses or a homeostasis was reached between phlorotannin production and removal by secretion and/or cross-linking on the cell wall. In order to go further on these eventualities, metabolic turn-overs occurring under grazing conditions can be monitored by measuring in situ rates of synthesis, polymerization, and turnover of extractable phlorotannins, as this has previously been done on tropical brown seaweeds by using a stable isotope labeling process [32]. Another possibility could be to carefully follow the variation of insoluble phlorotannin contents by using an appropriate extraction protocol [33].

### 3.2. Gene Expression

In order to understand more thoroughly the response of *F. vesiculosus* during the first hours of grazing, a qRT-PCR approach was deployed.

Results showed that the expression of the gene encoding HSP70 was not affected by grazing in opposition to previous results obtained when investigating UV-B stress exposure in *F. vesiculosus* [28]. Indeed, this *hsp70* gene has also been reported to overexpress in *Ulva* under high UV conditions, a response that could result from the perception of molecular thermic agitation caused by UV radiations [34]. Heat shock proteins are mainly involved in thermal stress responses but can also be related to other abiotic stresses [35,36]. Whilst our study shows that localized mechanical wounding by grazers does not modify *hsp70* gene expression, other authors have reported that this gene can be regulated by applying oligo-alginates in the culture media of *L. digitata* sporophytes, thereby chemically mimicking a biotic stress on the whole algae [37].

Regarding the *cyp450* gene, we observed that grazing lead to an upregulation at 24 h followed by a downregulation at 72 h. CYP450 proteins are represented by a large family of enzymes with distinct and complex functions involving multiple metabolic pathways [38,39]. In our study, the targeted *cyp450* gene was phylogenetically close to the *CYP5164B1*, a gene coding an epoxyalcohol synthase and involved in the production of oxylipin in the brown alga *Ectocarpus siliculosus* [40]. This gene has been shown to be up-regulated under hypersaline stress and down-regulated under oxidative stress applications [41], thereby indicating a strong reactivity in multiple stress responses. However, a phylogenetic study (Appendix A), showed that the *cyp450* gene targeted in our study did not branch directly with *CYP5164B1*. This observation could be interpreted as a duplication of this gene in Fucales which does not occur in Ectocarpales. This duplication could have led the gene to either preserve its initial function or evolve towards the production of another enzyme. The characterization of this enzyme by using specific enzymatic tools, as it has been described when characterizing CYP-74 from *Ectocarpus siliculosus* [40], should permit to identify its biochemical activity.

The *vbpo* gene targeted in the present study was overexpressed only 6 h after contact between *F. vesiculosus* and *L. littorea*. A previous study showed that the same *vbpo* gene from *F. vesiculosus* was upregulated 2.8 times after 48 h of grazing by *L. obtusata* and 4.8 times after 4 days, but with no differential expression on day 1 and 3 [19]. Although *vbpo* gene expression can be activated depending on the type of herbivore attack, we must keep in mind that other external parameters, specific to each study, can also be involved in the response. The vBPO enzyme was selected in this present study due to its potential role in the scavenging of Reactive Oxygen Species (ROS) and in the cross-linking of phenols, possibly phlorotannins, with cell wall alginates and phlorotannins [5,6,42,43,44]. However, the occurrence of several multigenic families of vBPOs has been reported in brown algae [45], which could imply multiple functional roles and maybe different local regulations according to the applied stress.

Interestingly the aryl sulfotransferase (*ast6*), the polyketide synthase (*pksIII*) and the *x22181* genes were all three simultaneously upregulated after 24 h of grazing, thereby suggesting a joint response to the grazing activity. The *pksIII* gene, encodes the enzyme for the synthesis of phloroglucinol as the precursor of phlorotannin chains, and the *x22181* gene is potentially involved in maintaining membrane integrity. The gene *ast6* encodes an enzyme involved in the sulfation of various compounds (including phenols), which is a chemical process able to facilitate their solubility [46]. Authors here assume that the stimulation of phlorotannin sulfation could probably be necessary in order to maintain their soluble status until the integration within the cell wall [2]. So far, no information is available regarding the sub-cellular localization of the PKSIII protein in the *Fucus* species. However, it would be of great interest to gain more knowledge in identifying the compartment where the phloroglucinol is produced and in investigating the possible link with the formation of physodes.

The dynamic response of the *x22181* gene could suggest a very important role in maintaining the membrane integrity of cells which are located in the vicinity of grazed tissues. However, further characterization of the corresponding protein would be required to better understand its induction. The apparent discrepancy in the timing of the overexpression of the *pksIII* gene, which correlates with the upregulation of the *x22181* gene, might be explained by a slower signal transduction that requires the gene-regulated synthesis of some metabolites such as oxylipins which were not targeted in the present study, but were shown to be important in the herbivore-induced responses in kelps [47].

The upregulation of the *pksIII* gene measured after 24 h grazing might have been triggered by the need to maintain a homeostasis of soluble phlorotannins, as shown through the concentrations measured along the kinetics during grazing. Interestingly, Flöthe et al. [23] recently showed through a global transcriptomic analysis that *F. vesiculosus* displayed multiple defense pulses in response to the periwinkle *Littorina obtusata* grazing. Such results hence suggest a high level of temporal variability in antiherbivory traits. However, it is important to keep in mind that this latter study monitored responses several days after grazing and short-term transcriptome changes were not monitored. However, among the 400 genes which were significantly up-/down-regulated after *I. baltica* grazing, genes involved in the photosynthesis were the mostly down-regulated after 15 and 18 days, while genes related to intracellular exchanges, secretion, vesicular transport and to the respiratory chain were on the other hand up-regulated. Such results indicate that grazing clearly affected the allocation of resources within the algae [23]. Unfortunately, this study did not measure the phenolic content of grazed tissues.

To conclude, our results are in agreement with previous studies showing the inducibility of the phlorotannin metabolism in response to grazing. Targett and Arnold [48] support the idea that herbivory has a significant and primary effect on the induction of phlorotannins in different species of *Fucus*. An increase of soluble phlorotannin concentrations in *F. vesiculosus* has also been reported after 3 days and even after longer periods (2 weeks) of grazing by *Littorina littorea* [19]. Similar results were also found when studying *F. distichus* grazed by *Littorina sitkana* [49]. In addition, the induction of phlorotannin was revealed by microscopic visualisation in tissues of the kelp *E. radiata* previously submitted to mechanical damaging [18]. Inducible defenses do not only reduce seaweed vulnerability to herbivore attack but can also have the potential to modify the feeding behavior of herbivores (repugnace) and competition among several herbivore species may be mediated by induced changes in seaweed traits [50,51].

## 4. Materials and Methods

### 4.1. Biological Materials and Experimental Design

*Fucus vesiculosus* thalli were freshly collected from the littoral zone of Kiel Fjord at Kiekut (54°26’54.8” N 9°52’21.9” E) and *Littorina littorea* was collected from a rocky shore at Mönkeberg (54°21’07.8” N 10°10’39.4” E). Induction experiments were run in a constant temperature chamber (15 °C) during August 2013 at the Helmholtz Center for Ocean Research (GEOMAR) in Kiel, Germany.

The experimental set-up consisted of a flow-through system with ambient filtered water (1.2 µm) pumped from the nearby Kiel Fjord. Water was stored in a tank (150 L) before supplying the 90 transparent plastic aquaria (2.9 L), with a permanent renewal of seawater based on a constant flow rate and bubbled air. Light was provided by fluorescent tubes (OSRAM FLUORA, Munich, Germany, L 36 W/77 25 × 1) above the aquaria. The total irradiance was 14.24 ± 0.04 W.m^−2^ with a light/dark period of 14/10 h. In order to prevent the escape of snails, each aquarium was covered by a metal grid. The acclimation of *F. vesiculosus* to the culture conditions lasted 1 week and *L. littorea* were assigned in an independent aquarium without food during 3 days before the induction experiment.

Thalli of *F. vesiculosus* were thoroughly washed with filtered seawater, weighed (in order to calculate the necessary amount of snails per g of fresh tissues) and assigned to aquaria. In each treatment aquarium, we applied 37 mg dry weight of snails, estimated from the shell diameter per g of algal fresh weight, using a previously determined regression formula between both measurements [52].
Dry weight [mg] = 6614 × e^(0.2232 × diameter[mm])

The incubation time was 3 weeks, with successive sampling points of five replicates for each condition.

### 4.2. RNA Extraction

The RNA extraction protocol was adapted from Apt and Grossman (1993) [53], Apt et al. (1995) [54] and Pearson et al. (2006) [55]. 50 mg dry weight (DW) of freeze-dried tissue were ground for 5 min at 6500 rpm at room temperature using a mixer-mill (Precellys 24, Bertin Technologies) in 2 mL Eppendorf tubes with ceramic beads supplied. Extraction buffer consisted of Tris-EDTA (100 mM Tris, 50 mM EDTA, pH 7.5, with 1.5 M NaCl and 2% CTAB). Immediately prior to extraction, 500 mM of DTT was added as antioxidant from a 1 M stock dissolved in water. Then, 1.5 mL of extraction buffer was added per 50 mg of dry weight tissue, and the suspension was mixed vigorously by vortexing. After 15 min of extraction on ice, the mixture was centrifuged at 10,000× *g* for 20 min at 4 °C. The supernatant solution was transferred to a new tube and 1 volume of chloroform:isoamyl alcohol (24:1 *v*/*v*) was added, vortexed vigorously and centrifuged at 10,000× *g* for 20 min at 4 °C. This second supernatant was then collected in a new tube and 0.2 volume of absolute ethanol was gently added and mixed by rocking the tube. Ethanol addition resulted in the precipitation of polysaccharides [56]. A second chloroform extraction was then carried out under the same conditions and the supernatant was carefully removed. RNA was precipitated with 0.4 volumes of 12 M LiCl in the presence of 1% (*v*/*v*) β-mercaptoethanol as antioxidant. Precipitation was performed overnight at −20 °C. The RNA was collected by centrifugation at 10,000× *g* for 30 min at 4 °C, the RNA pellet was dried up (10–20 min on ice) and then re-suspended in 500 µL RNase-free water. The RNA was extracted with an equal volume of phenol-chloroform:isoamylalcohol (24:1 *v*/*v*) (1:1) and centrifuged at 10,000× *g* for 15 min at 4 °C. The resulting pellet was washed by 1 volume of chloroform:isoamyl alcohol (24:1 *v*/*v*) in order to remove the phenol. RNA was re-precipitated with 2 volumes ethanol, 0.3 M sodium acetate and re-suspended in 20 µL of sterile H_2_O. The RNA was treated with RNAse-free DNAse-I according to the manufacturer’s instructions (Qiagen, Hilden, Germany) to remove any contaminating DNA. The purity and concentration of RNA was measured by a NanoDrop 2000 spectrophotometer (ThermoScientific, Waltham, Massachusetts, USA) (A260/280), RNA integrity was assessed by agarose gel electrophoresis. RNA was stored at −80 °C.

### 4.3. Quantitative Real-Time PCR (qRT-PCR)

The reverse transcription (RT) was performed on 250 ng total RNA, with 1 µL of oligo (dT) 18 (100µM). Denaturation was performed at 70 °C for 5 min. Then the master mixture containing 5 µL Improm II Buffer 5X (Promega, Madison, Wisconsin, USA), 4 µL MgCl2 (25 mM), 1 µL dNTP mix (10 mM), 0.5 µL RNAsine, 2.5 µL Nuclease free water and 1 µL of ImProm-II™ Reverse Transcriptase (Promega) was added. The reaction mixture was pre-incubated at 25 °C for 5 min, the extension was carried out at 42 °C for 60 min, and the RT was inactivated by heating at 70 °C for 15 min. qRT-PCR was carried out using the LightCycler^®^ 480 multiwell plate 96, on a LightCycler^®^ 480 Real-Time PCR System (Roche Diagnostics, Mannheim, Germany) in three technical replicates, using 5 µL of the LightCycler^®^ 480 SYBR Green Master mix (Roche Diagnostics, Mannheim, Germany) with 2.5 µL of cDNA (1 ng.µL^−1^) or genomic DNA (gDNA) for quantification, 0.5 µL of each primer (10 µM) and 1.5 µL water for a final volume of 10 µL.

The cycling program for PCR quantification was as follows: 5 min denaturation at 95 °C, followed by 45 cycles of 95 °C for 10 s, 60 °C for 15 s, and 72 °C for 15 s. Melting curves were also programmed at 97 °C in order to control the risk of multiple amplifications.

### 4.4. Gene Expression Study

Primer pairs were designed using Primer3plus (http://primer3plus.com/cgi-bin/dev/primer3plus.cgi, accessed on 15 February 2011).

The design of a set of primers (Table 1) was based on EST clones from cDNA libraries representing of desiccated *F. vesiculosus* and *F. serratus* [27], a *F.vesiculosus* genome [57] and a microarray study [25] accessible through GEO series accession number GSE47975 (http://www.ncbi.nlm.nih.gov/geo/query/acc.cgi?acc=GSE47975, accessed on 20 February 2011). Relative gene expression was analyzed with LightCycler^®^ 480 software (Roche Diagnostics, Mannheim, Germany), using the 2^-∆CT method by using as references genes *tua* (Alpha tubulin) and *ef1α* (Translation elongation factor 1 alpha) [27].

### 4.5. CYP450 Phylogenetic Analysis

The CYP450 protein coding sequence of *F. vesiculosus*, used in RT-qPCR experiments of this study, was searched for brown algal homologues by a tblastn approach against the genomic (WGS) and transcriptomic (TSA) NCBI public databases. Its protein sequence is identical to that encoded by JACAZD010001283.1*. Ectocarpus siliculosus* and *Saccharina japonica* sequences were retrieved from Teng et al., 2019 [41]. A total set of 32 protein sequences were loaded into a NGPhylogeny.fr “à la carte” pipeline (https://ngphylogeny.fr/, accessed on 10 January 2021). Proteins were first aligned with MAFFT 7.407_1 program leading to a superposition of 666 initial positions. The alignment was then curated by BMGE 1.12_1 according to default parameters (Maximum entropy threshold = 0.5; Gap Rate cut-off = 0.5; Minimum Block Size = 5). 279 informative positions were retained by the tool and selected to build the phylogenetic tree. The maximum likelihood PhyML+SMS 1.8.1_1 method was chosen with standard criterions (Statistical criterion to select the model = AIC; Tree topology search = SPR; Branch support = SH-like aLTR). The best substitution model (SMS) was found to be WAG+G+I and was applied to infer the tree modelisation. Finally, the Newick display of the tree was rendered as a dendrogram with the iTOL v5.7 viewer (Biobyte solutions, Heidelberg, Germany) [58].

### 4.6. Phlorotannin Extraction

Phlorotannins were extracted from 100 mg dry weight (DW) of freeze-dried tissue. Tissues were ground in 2 mL Eppendorf tubes with metal beads during 5 min at 6500 rpm at room temperature, using a mixer-mill (Precellys 24, Bertin Technologies, Montigny-le-Bretonneux, France). Extraction was performed 3 times successively on the obtained tissue powder with methanol:water (80:20) at pH 4.3 in dark at 40 °C during 30 min with agitation in a thermomixer (Eppendorf, Hambourg, Germany). The extract was centrifuged 10 min at 10,000× *g* and the supernatant was removed. Methanol was evaporated in a speed-vacuum concentrator miVac Duo Concentrator (miVac, Genevac Limited, Ipswitch, UK) at 40 °C and the total extract was lyophilized and weighed.

### 4.7. Quantification of Soluble Phlorotannins

The quantification of total soluble phorotannins in the extracts was performed using the adapted Folin–Ciocalteu method [59] with phloroglucinol used as a standard (Sigma, Saint-Louis, Missouri, USA). Each sample was re-suspended in 1 mL methanol:water (80:20) at pH 4.3 and diluted to reach a concentration of 1 mg.mL^−1^. Quantification was carried out using multiwell plates (Nunc UV-Star 96 wells). The reaction was performed with 20 µL of extract (1 mg.mL^−1^), 40 µL of Na_2_CO_3_ 20%, 130 µL milliQ water and 10 µL Folin-Ciocalteu reagent (Sigma). The reaction was heated at 70 °C for 10 min with a cover in a thermocycler and the absorbance of the solutions was then measured at 750 nm in multiwell plates on a Safire2Tecan Multi-detection Microplate reader (ThermoScientific, Waltham, MA, USA).

### 4.8. Statistical Analyses

All values obtained under the different experiments and conditions were analyzed using two-way analysis of variance (two-way ANOVA *p* < 0.05, *p* < 0.1). Mean comparisons were made using multiple comparisons of means Tukey contrasts test or estimated marginal means (emmeans) test with significant differences reported at *p* < 0.05. All statistical analyses were done using R version 4.0.2 (R Foundation for Statistical Computing, Vienna, Austria) with R package [60].

## 5. Conclusions

Combining our results points out the following sequence of events involved in the metabolism of phlorotannin in *F. vesiculosus* grazed by *L. littorea*: (i) mobilization of the pool of phloroglucinol malonyl-CoA precursors initially occurring in the cells to activate the pool of pksIII enzyme as a quick response to grazing during the very first hours of a grazer intrusion, as suggested by the increase of soluble phenol contents, (ii) local induction and increase of some oxidative mechanisms involving enzymes like vBPO, (iii) constant level of soluble phenol level after 24 h which was maintained by the overexpression of *pksIII* and *ast6* genes to compensate possible losses by exudation and/or phenol cross-linking in the cell walls. Indeed, as suggested by Koivikko et al. [61], the cross-linking of phlorotannins is liable to modify its chemical properties, which can hence impact the quantification process. Moreover, phlorotannins can also diffuse through the cell wall by exudation, thereby taking part in a global turnover of the compound [32].

In contrast to our previous study which showed an adaptive constitutive mechanism of phlorotannin accumulation in *F.vesiculosus* to prevent excess UV-B irradiation [28], this present study supports the idea that phlorotannins can also provide an inducible chemical defense in the presence of herbivores [18,26]. Phlorotannins are upregulated in response to periwinkle grazing and this result is therefore in agreement with their important role when considering interactions between brown macroalgae and herbivores. However, more information on their biosynthesis pathway and chemical modifications is necessary to fully understand these mechanisms. To that effect, it would be interesting to characterize additional genes and their functions and also investigate the activation of specific genes in response to grazing in parallel to the characterization and subcellular localization of phlorotannins.

Brown algal phlorotannins have been reported to potentially provide a variety of health benefits [62]. Indeed, these compounds have previously shown promising activities in terms of antioxidative, antidiabetic, anti-inflammatory and antitumor activities, among much more [63,64,65,66]. By using multidisciplinary approaches, the characterization of mechanisms involving phlorotannin synthesis and derivatives could promote biotechnological developments which in turn could be helpful for providing drug candidates for pharmacological applications.

## Figures and Tables

**Figure 1 marinedrugs-19-00185-f001:**
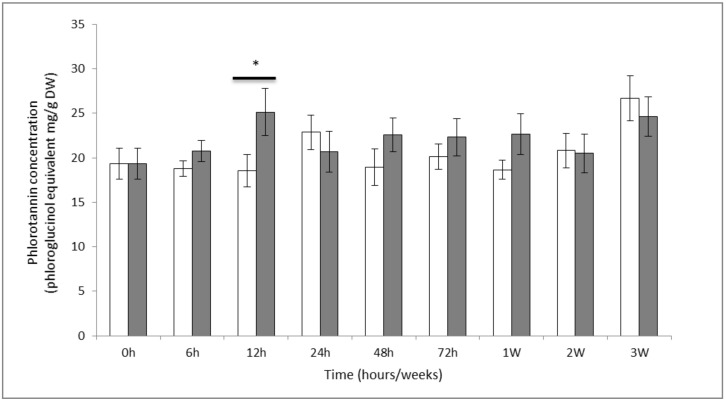
Quantification of total soluble phenol contents (mg equivalent phloroglucinol.g^−1^ DW) in methanolic extract in control condition (white square) and exposed to grazing (black square). Values represent means of five independent replicates and bars represent the SE. Asterisks * indicate significant differences between control and grazing treatment (estimated marginal means, *p* value < 0.05).

**Figure 2 marinedrugs-19-00185-f002:**
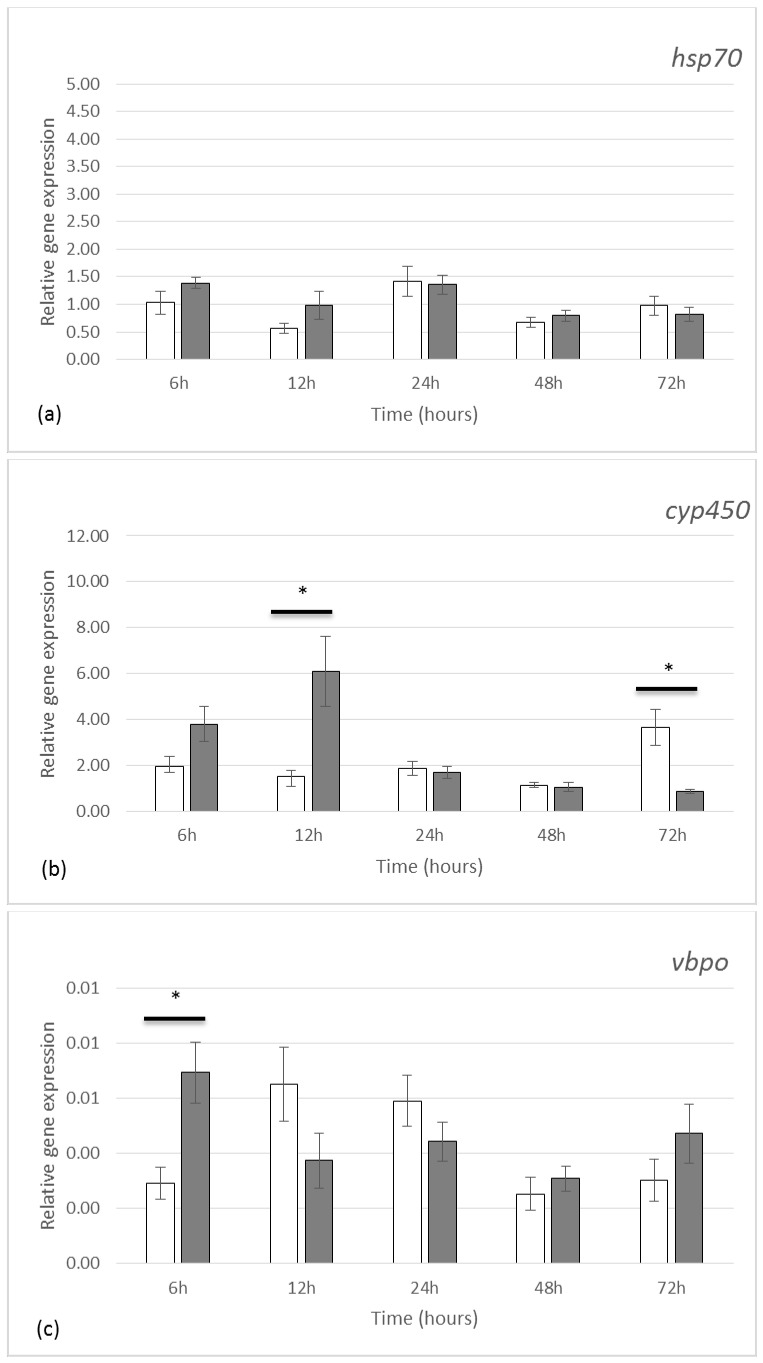
Relative gene expression in *Fucus vesiculosus* in controlled condition (white square) and exposed to grazing (grey square) during 72 h are presented for (**a**) *hsp70*, (**b**) *cyp450*, (**c**) *vbpo*, (**d**) *ast6*, (**e**) *pksIII*, (**f**) *x22181*. The expression of a gene is normalized to the geometric mean of the expression of 2 reference genes (*ef1α*, *tua*) [27] in the same algal sample at each time point. Values represent means of five independent replicates and bars represent the SE. Asterisks * indicate significant difference between the control and the grazing treatment (Tukey-test, *p* value < 0.05).

**Table 1 marinedrugs-19-00185-t001:** Primers used for the qRT-PCR analysis on control and grazing conditions.

*Gene*	Forward	Reverse	Amplicon Length (bp)	Tm °C	Accession Number	References
***ef1.alpha***	TGCGTACAATCGCATTCG	CGAAACATGAAGGACAGTTGC	198	58	GH706096	EST (*Pearson et al. 2010*)
***tua***	GTCACACCGATGTAGAGGA	GGCTTCCAGACAATTACCC	96	58	GH702736	EST (*Pearson et al. 2010*)
***pks***	TTGCACGTATGTCTCTGTTGC	GCGCGAATAACCTGATGG	135	60	GH706741	EST (*Pearson et al. 2010*)
***vbpo***	CCAAGGCGTCGAGTCATATC	GCACTTACTGCAATCCAATGTAC	129	59	CUST_1779_PI408257168	Microarray accession number GSE47975 *(Flöthe et al 2014 Eur J.Phycology)*
***hsp70***	AGATCGAGGAGATTGACTAGATGG	CGACTTGCATCACACATATCG	161	60	GH704979	EST (*Pearson et al. 2010*)
***ast6***	GACCCTTCCCTGATCTTCC	CCAGATGCGGTCATTTCAC	83	59	JACAZD010018410.1 (19554-20266)	*Fucus vesiculosus* genome
***x22181***	TGGTCGAGACGGAGGAAG	TGCACTTCAAGCTATTACTCTTGC	132	60	CUST_44291_PI408257168	Microarray accession number GSE47975 *(Flöthe et al 2014 Eur J.Phycology)*
***cyp450***	TAACGACATGGCTCAAATCAC	ACACAACAAACACCCACAC	84	59	CUST_44854_PI408257168	Microarray accession number GSE47975 *(Flöthe et al 2014 Eur J.Phycology)*

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
