# Peer review of "Induction of Phlorotannins and Gene Expression in the Brown Macroalga Fucus vesiculosus in Response to the Herbivore Littorina littorea"

_marinedrugs, 2021, doi:10.3390/md19040185_

Round 1

Reviewer 1 Report

This study is of value in providing data on the time course of induction of specific genes over the first three days after the initiation of grazing. (The Flöthe et al. papers cited provide much greater coverage but not the same temporal resolution provided here.) However, the data on phlorotannins, and in particular the purported links between the genetics and phlorotannin assays, are weak. The abstract (line 25) says “After only 12 hours…” phlorotannins are induced when in reality it is “Only after 12 hours…” where they were significantly. There is a trend that looks like induction at before and several intervals thereafter but not significantly. Perhaps the sample size did not provide enough statistical power to identify what was really happening, but the data are what they are. This is particularly in issue with interpreting the pksIII data (which is only significantly upregulated later, albeit with the same potential power caveat) since the authors’ postulate (with reason) that it is involved in the production of phlorotannins. Their hand-waving about using up pools of phlorotannins prior to that seems weak since soluble pools are what is being measured. It also seems to make an attempt (lines 137-138) to explain the weak response in terms of using much more tissue for the phlorotannins than was actually grazed. That, however, is pretty standard and what has been done in other papers that have shown induction of phlorotannins. Overall, while the paper does look at stress response, I think the final statement in the abstract about this telling us much of value about phlorotannin induction dynamics is unwarranted.

So too is the statement in the conclusions (line 362-363) that this provides evidence that phlorotannins provide defense against herbivores. This study did not look at defenses against herbivores. Although other studies have shown that unpalatability of F. vesiculosus increases in response to grazing of other Littorina species, no such data are provided here. As the authors do a good job of introducing, phlorotannins have lots of secondary and primary roles in brown algae including a primary role in cell wall formation/strengthening and that it is not always correlated with increases in unpalatability to herbivores. It is long past (15 years or more) since one could get away with assuming that an increase in phlorotannins causes herbivore deterrence. The authors cite work of their own in review (ref. 20; does journal policy allow citation of submitted papers??) that claims L. littorea grazing reduces palatability but no details of how the experimental design or response timing correspond to those here. It also says that “phloroglucinol” is impacting the grazing which I certainly hope is a mistake and that they mean “phlorotannins.” Even the smallest phlorotannin size classes are made up of a great many phloroglucinol subunits and I am not aware of any prior evidence for a defensive or other role for the monomers themselves.

Reference 3 (Koivikko’s dissertation) citation is truncated.

There are a few places in the manuscript where the word choice or order is not that of a native English speaker but I presume that those would be corrected by a copy editor if the paper is accepted.

Author Response

This study is of value in providing data on the time course of induction of specific genes over the first three days after the initiation of grazing. (The Flöthe et al. papers cited provide much greater coverage but not the same temporal resolution provided here.) However, the data on phlorotannins, and in particular the purported links between the genetics and phlorotannin assays, are weak. The abstract (line 25) says “After only 12 hours…” phlorotannins are induced when in reality it is “Only after 12 hours…” where they were significantly. Sentence has been corrected.

There is a trend that looks like induction at before and several intervals thereafter but not significantly. Perhaps the sample size did not provide enough statistical power to identify what was really happening, but the data are what they are. This is particularly in issue with interpreting the pksIII data (which is only significantly upregulated later, albeit with the same potential power caveat) since the authors’ postulate (with reason) that it is involved in the production of phlorotannins. Their hand-waving about using up pools of phlorotannins prior to that seems weak since soluble pools are what is being measured. It also seems to make an attempt (lines 137-138) to explain the weak response in terms of using much more tissue for the phlorotannins than was actually grazed. That, however, is pretty standard and what has been done in other papers that have shown induction of phlorotannins. Overall, while the paper does look at stress response, I think the final statement in the abstract about this telling us much of value about phlorotannin induction dynamics is unwarranted.

We modified abstract by “This study supports the hypothesis that phlorotannins are able to provide an inducible chemical defense under grazing activity which is regulated at different stages of the stress response.”

So too is the statement in the conclusions (line 362-363) that this provides evidence that phlorotannins provide defense against herbivores. This study did not look at defenses against herbivores. Although other studies have shown that unpalatability of F. vesiculosus increases in response to grazing of other Littorina species, no such data are provided here. As the authors do a good job of introducing, phlorotannins have lots of secondary and primary roles in brown algae including a primary role in cell wall formation/strengthening and that it is not always correlated with increases in unpalatability to herbivores. It is long past (15 years or more) since one could get away with assuming that an increase in phlorotannins causes herbivore deterrence. Modifications has been done

The authors cite work of their own in review (ref. 20; does journal policy allow citation of submitted papers??) that claims L. littorea grazing reduces palatability but no details of how the experimental design or response timing correspond to those here. It also says that “phloroglucinol” is impacting the grazing which I certainly hope is a mistake and that they mean “phlorotannins.” Even the smallest phlorotannin size classes are made up of a great many phloroglucinol subunits and I am not aware of any prior evidence for a defensive or other role for the monomers themselves.> Regarding comments of reviewers on this article, we prefer to delete this reference.

Reference 3 (Koivikko’s dissertation) citation is truncated.- Done

There are a few places in the manuscript where the word choice or order is not that of a native English speaker but I presume that those would be corrected by a copy editor if the paper is accepted. Correction of English has been done

Reviewer 2 Report

The manuscript entitled “Induction of phlorotannins and gene expression in the brown macroalga Fucus vesiculosus in response to the herbivore Litto-rina littorea” comprises the necessary elements of scientific novelty.

Minor comments:

  • Line 41-46 a) this part should have been written at beginning of the introduction (reframe the sentence).
  • Line 72-78: a) The information in this paragraph has no proper link in a sentence- rewrite. b) Be specific to study?
  • The introduction part needs to be improved. Can you please mention the source of Phlorotannins in line 49?
  • Line 93- What basis the genes were selected??
  • Line 123- How about the internal control or reference gene –data or details??
  • Line 225 - Mönkeberg (N54.352866°, E10.177580°)...- Typo error
  • Please improve the Figure 1 resolution, it looks blurred.
  • The discussion part needs to be rewritten and please discuss your findings with recent studies.
  • This study revealed new information about how the expression of genes in vesiculosus. Moreover, the gene expression pattern was monitored over different time intervals. Nevertheless, the RT-qPCR analyses are a) neither comprehensive nor b) do the data fit together (with respect to a focused topic), and are the results further corroborated by any other method.
  • The outcome of the research in the abstract is not the same as that at the end of the Conclusion section.

I would recommend the publication of this manuscript after addressing minor changes.

Author Response

Minor comments:

  • Line 41-46 a) this part should have been written at beginning of the introduction (reframe the sentence). We make the choice to describe firstly chemical composition of phlorotannins before their localization.
  • Line 72-78: a) The information in this paragraph has no proper link in a sentence- rewrite. b) Be specific to study? Modifications has been done
  • The introduction part needs to be improved. Can you please mention the source of Phlorotannins in line 49?ok
  • Line 93- What basis the genes were selected?? Genes were selected regarding previous studies, we added references at this line.
  • Line 123- How about the internal control or reference gene –data or details?? EF1alpha and TUA were used as reference genes –

Reference Pearson, G.; Hoarau, G.; Lago-Leston, A.; Coyer, J.; Kube, M.; Reinhardt, R.; Henckel, K.; Serrão, E.; Corre, E.; Olsen, J. An expressed sequence tag analysis of the intertidal brown seaweeds Fucus serratus(L.) and F. vesiculosus (L.) (Heterokontophyta, Phaeophyceae) in response to abiotic stressors. Mar. Biotechnol. 2010, 12, 195–213.

  • Line 225 - Mönkeberg (N54.352866°, E10.177580°)...- Typo error ok
  • Please improve the Figure 1 resolution, it looks blurred. ok
  • The discussion part needs to be rewritten and please discuss your findings with recent studies. Modifications has been done
  • This study revealed new information about how the expression of genes in vesiculosus. Moreover, the gene expression pattern was monitored over different time intervals. Nevertheless, the RT-qPCR analyses are a) neither comprehensive nor b) do the data fit together (with respect to a focused topic), and are the results further corroborated by any other method. This comment is not really clear: could you please explain us what is incomprehensive? We did not corroborate our results by another method
  • The outcome of the research in the abstract is not the same as that at the end of the Conclusion section. Modifications has been done in the abstract

I would recommend the publication of this manuscript after addressing minor changes.

Round 2

Reviewer 1 Report

The authors have made reasonable revisions in response to my comments on the first version.

Author Response

Many thanks for your constructive comments they have allowed us to improve this manuscript.

Reviewer 2 Report

No further comments.

Author Response

(The authors gave the same response as above.)
